# Molecular Diagnosis of Human Monkeypox Virus during 2022–23 Outbreak: Preliminary Evaluation of Novel Real-Time Qualitative PCR Assays

**DOI:** 10.3390/microorganisms12040664

**Published:** 2024-03-27

**Authors:** Vanessa De Pace, Bianca Bruzzone, Valentina Ricucci, Alexander Domnich, Giulia Guarona, Giada Garzillo, Rexhina Qosja, Giulia Ciccarese, Antonio Di Biagio, Andrea Orsi, Giancarlo Icardi

**Affiliations:** 1Hygiene Unit, San Martino Policlinico Hospital-IRCCS for Oncology and Neurosciences, 16132 Genoa, Italy; bianca.bruzzone@hsanmartino.it (B.B.); valentina.ricucci@hsanmartino.it (V.R.); alexander.domnich@hsanmartino.it (A.D.); rexhina.qosja@hsanmartino.it (R.Q.); andrea.orsi@unige.it (A.O.); icardi@unige.it (G.I.); 2Laboratorio di Riferimento Regionale per le Emergenze di Sanità Pubblica (LaRESP), 16132 Genoa, Italy; giuly.guarons@outlook.it (G.G.); giadagarzillo@gmail.com (G.G.); 3Department of Health Sciences, University of Genoa, 16132 Genoa, Italy; 4Dermatology Unit, San Martino Policlinico Hospital-IRCCS for Oncology and Neurosciences, 16132 Genoa, Italy; giulia.ciccarese@unifg.it; 5Infectious Disease Unit, San Martino Policlinico Hospital-IRCCS for Oncology and Neurosciences, 16132 Genoa, Italy; antonio.dibiagio@hsanmartino.it

**Keywords:** monkeypox virus, diagnosis, *real-time* PCR

## Abstract

In 2022–23, the human monkeypox virus (MPXV) caused a global outbreak in several non-endemic countries. Here, we evaluated the diagnostic performance of four *real-time* qualitative PCR assays for the laboratory diagnosis of mpox (monkeypox) monkeypox disease. From July to August 2022, 27 positive and 10 negative specimens (lesion, crust and exudate swabs) were tested in the laboratory of the Hygiene Unit of the San Martino Hospital (Genoa, Italy) by using home-made real-time PCR to detect MPXV generic G2R_G DNA. According to the manufacturer’s instructions, we also retrospectively analyzed these specimens using RealCycler MONK-UX/-GX (Progenie Molecular), STANDARD M10 MPX/OPX (SD Biosensor), Novaplex MPXV (Seegene Inc.) and RealStar Orthopoxvirus PCR Kit 1.0 (Altona Diagnostics) assays, recognized as research-use-only tests. The diagnostic accuracy and sensitivity of these assays ranged from 97.3% (95% CI: 86.2–99.5%) to 100% (95% CI: 90.6–100%) and 96.3% (95% CI: 81.72–99.34%) to 100% (95% CI: 72.2–100%), respectively. The RealCycler MONK-UX and STANDARD M10 MPX/OPX did not detect one positive sample with a cycle threshold of 36. The overall specificity was 100% (95% CI: 72.2–100%), and Cohen’s Kappa values ranged from 1 (95% CI: 0.67–1) to 0.93 (95% CI: 0.61–1). As they are highly accurate, reliable and user-friendly, these tests should be recommended for the routine or rapid laboratory discrimination of mpox from other rash illnesses.

## 1. Introduction

Monkeypox virus (MPXV) is a cause of a zoonotic disease called mpox (previously known as monkeypox), which continues to be transmitted from the endemic region of Western and Central Africa to non-endemic countries, including Europe and the United States [1]. The 2022–23 mpox outbreak was declared a public health emergency of international concern (PHEIC) by the World Health Organization (WHO) since early May 2022 to May 2023 [2].

As of 31 January 2024, there were a total of 9,392,193,921 laboratory-confirmed cases and 179 deaths. In this WHO global overview update, the number of monthly reported new cases increased by 29.7% compared to the previous month. The region of the Americas (52.1%) and the European Region (22.1%) already accounted for the majority of cases reported [1]. Therefore, in most WHO regions, the human-to-human transmission of MPXV continues.

MPXV is related to variola virus (VARV), the cause of smallpox, and both are members of the genus Orthopoxvirus. MPXV has an enveloped double-stranded DNA genome and belongs to the genus *Orthopoxvirus* of the family *Poxviridae* [3,4,5]. The MPXV genome contains a conserved central region (OPG048 to OPG151) and variable terminal regions that consist of inverted terminal repeats (ITRs) [6,7]. Genes for genome replication, essential enzymes and structural proteins are located in this central region. Virulence and host range genes are encoded by the variable terminal regions [6]. In 1959, the first report of an MPXV diagnosis involving monkeys was described, and this later resulted in a smallpox virus-like outbreak with less severe symptoms and outcomes [8]. Since the first human case of MPXV in the Democratic Republic of Congo in 1970, several host reservoirs have been identified, including squirrels, rats, dormice, non-human primates and other species [9,10,11,12]. Mpox is transmitted from animals to humans or from human to human by direct contact with skin or mucosal lesions and respiratory droplets from infected individuals [4]. During the 2022–23 outbreak, bisexuals and men who have sex with men were the most affected, suggesting that the viral transmission is facilitated by close and intimate physical contact [1]. As shown by the WHO global data report, 96.4% of cases with available data were males, and their median age was 34 years (IQR: 29–41) [1]. Transmission can also occur via indirect contact (fomites, biological fluids) or vertically (placenta). Systemic symptoms (such as fever, headache, lymphadenopathy, asthenia, myalgia and back pain) and cutaneous rash are common clinical signs of the infection, and these symptoms are usually identified 6 to 21 days after incubation [2]. However, mpox disease is much less severe than smallpox, with a milder rash, except for pregnancy or immunodeficiency conditions, where MPXV can cause severe disease [13].

Following the genomic analysis of the terminal regions encoding host response modifier proteins, two major clades of mpox have been defined: West Africa (WA MPXV or clade 2) and Congo Basin (CB MPXV or clade 1). Severe and more transmissible cases are associated with CB MPXV, whereas low virulence is associated with WAMPXV [13]. Clade I causes a smallpox-like illness and has a fatality rate of up to 10% in unvaccinated populations. A significant decrease in T-cell-mediated cytokine production has been observed in the CB MPXV infected cases [13]. Therefore, this virulent clade down-regulates host responses, preventing T cell receptor-mediated T cell activation [13].

The demand for laboratory testing of mpox has increased since the first non-endemic case was reported in May 2022. This was a British traveler returning from Nigeria, a known endemic area. Testing for mpox is recommended for patients presenting with a suggestive rash and individuals reporting close contact with an infected case [14]. Mpox virus is diagnosed by real-time qualitative polymerase chain reaction (PCR) on swabs taken from mucosal or skin lesions. Blood samples are not recommended for molecular testing because viral shedding in the bloodstream occurs during the initial phase of infection, usually in the prodromal period and before skin lesions appear. Likewise, antigen and antibody assays may not be useful as laboratory procedures for mpox because orthopoxviruses are serologically cross-reactive [15].

The aim of this study was to evaluate the diagnostic performance of four commercial real-time PCR molecular assays in comparison with an in-house real-time PCR described by Li and colleagues for the diagnosis of mpox [16].

## 2. Materials and Methods

This clinical validation study was performed retrospectively on a collection of 37 clinical skin specimens (*n* = 27 true positive samples and *n* = 10 true negative samples) at the Hygiene Unit Laboratory of the San Martino Hospital (Genoa, Italy), which is the regional reference laboratory for emerging infectious diseases. Swabs of the lesion surface, exudate or lesion crusts were collected on virus transport and preservation medium (non-inactivated) (Biocomma Limited, Shenzhen, China) between 8 July 2022 and 5 August 2022.

### 2.1. Nucleic Acid Extraction

DNA extraction was performed automatically on a MagCore^®^ Super instrument (RBC Bioscience Corp., New Taipei, Taiwan) according to the manufacturer’s instructions. Nucleic acid was isolated from an initial volume of 400 µL and purified with an elution volume of 60 µL. Samples were handled in the BSL-2 laboratory facilities, as per the Centers for Disease Control and Prevention (CDC) recommendations [14].

### 2.2. In-House MPXV PCR Assay

MPXV generic G2R_G qualitative real-time PCR was performed on a CFX96™ instrument (BioRad Laboratories, Hercules, CA, USA) using the following thermal cycling profile: 1 cycle at 95 °C for 6 min and 45 cycles at 95 °C for 5 s and 60 °C for 20 s.

Primers and probe-targeting MPXV generic G2R_G were synthesized by TIB Molbiol (Genova, Italy), as follows: forward primer (5-GGAAAATGTAAAGACAACGAATACAG), reverse primer (5-GCTATCACATAATCTGGAAGCGTA) and probe sequence (5 FAM-AAGCCGTAATCTATGTTGTCTATCGTGTCC-3 BHQ1). G2R_G was designed within the TNF receptor gene [16]. The internal control (IC) was the human Ribonuclease P (RNAseP) gene. RNAseP primers and probe sequences were the following: forward primer (5-AGATTTGGACCTGCGAGCG), reverse primer (GAGCGGCTGTCTCCACAAGT) and probe sequence (5-6FAM-TTCTGACCTGAAGGCTCTGCGCG--BHQ1).

The total reaction volume was 20 µL (2 µL of template DNA/human internal control and 18 µL master mix). The master mix contained 5.5 µL of nuclease free water, 10 µL of EXPRESS qPCR Supermix (Invitrogen-ThermoFisher Scientific, Waltham, MA, USA), 1 µL each of MPXV generic G2R_G/RNAseP forward and reverse primers and 0.5 µL of G2R_G/RNAseP probes.

Each PCR assay was considered valid if the RNAseP gene was amplified with a cycle threshold (Ct) value ≤ 35. Positive results for the generic MPXV G2R_G gene were defined with the Ct values ≤ 35; Ct values between 36 and 40 were considered weakly positive.

### 2.3. Research-Use-Only (RUO) Assays for the Molecular Diagnosis of MPXV

Four research-use-only (RUO) molecular assays were evaluated in this study: Novaplex MPXV Assay (Seegene Inc.; Seoul, Republic of Korea); STANDARD M10 MPX/OPX (SD Biosensor, Suwon, Republic of Korea); RealCycler MONK-UX/MONK-GX v.2 (Progenie Molecular, Valencia, Spain); RealStar Orthopoxvirus PCR Kit 1.0 (Altona Diagnostics, Hamburg, Germany). These molecular tests were performed on specimens according to the manufacturer’s instructions. All samples were extracted on the same day of collection, and each template was stored at −20 °C until the amplification test.

#### 2.3.1. Novaplex MPXV Assay

It is a single-tube real-time PCR test for the detection of an MPXV target gene, including West African and Congo Basin clades. The type of validated specimen was not specified in the instructions for use. The detection of R1R viral amplicons, as part of the TNF receptor gene, is detected in FAM and IC RNase P is detected in Quasar 670 or CY5. RT-PCR is performed using the CFX96™ instrument with a thermal profile that takes approximately 40 min. Results were analyzed using the Seegene Viewer for RUO V3.

#### 2.3.2. STANDARD M10 MPX/OPX

STANDARD M10 MPX/OPX is a multiplex real-time PCR test intended for the qualitative detection of MPXV DNA in skin lesions, serum/plasma/whole blood and nasopharyngeal/oropharyngeal swabs collected from patients with suspected mpox. This procedure was performed on an automated STANDARD M10 system with a run time of 59 min. The MPXV G2R target (TNF receptor gene) can be distinguished from the Orthopoxvirus (OPX) target using a disposable cartridge containing primers, probes, IC and other real-time PCR reagents. MPXV G2R, OPX E9L and exogenous internal control targets are detected using the FAM, HEX and Cy5 fluorescent channels, respectively. In addition, this automated molecular method discriminates WA MPXV from CB MPXV using primers and probes for the intergenic regions, OPG181-OPG185 and F3L-F4L, respectively. Results are automatically interpreted by the STANDARD M10 Console.

#### 2.3.3. RealCycler MONK-UX/MONK-GX v.2

This commercial RUO kit from Progenie Molecular allows for the detection of MPXV DNA in pustule swabs and is based on ready-to-use reagents for real-time PCR. The viral TNF receptor gene and competitive heterologous IC targets are amplified using reverse and forward primers labeled with FAM and HEX probes, respectively. Real-time PCR was performed on the CFX96™ instrument with an amplification protocol of approximately 58 min. Both WA and CB MPXV can be detected by the RealCycler MONK-UX/MONK-GX v.2. BioVisor Pro-Q is recommended for the data interpretation of the samples and controls.

#### 2.3.4. RealStar Orthopoxvirus PCR Kit 1.0

The RealStar molecular system detects and discriminates the non-variola *Orthopoxvirus* species, including cowpox virus, MPXV, raccoonpox virus, camelpox virus and vaccinia virus, from the variola virus-specific DNA. The manufacturer did not specify the types of samples for which the assay may be used; specific sequences of primers and probes for non-variola orthopoxvirus targets and heterologous IC were also unavailable. The IC OPXV and VARV genes and IC amplicons are labeled with FAM, Cy5 and HEX fluorescent reporters, respectively. This molecular procedure is performed on the CFX96™ instrument and is characterized by a thermal profile of approximately 58 min.

### 2.4. Data Analysis

The clinical validation of these RUO MPXV molecular assays was performed by calculating the overall diagnostic accuracy, Cohen’s Kappa, diagnostic sensitivity and specificity, positive predictive (PPV) and negative predictive (NPV) values with the corresponding 95% confidence intervals (CIs). These parameters were estimated using the open-source Epidemiologic Statistics for Public Health (OpenEpi, https://www.openepi.com/, accessed on 1 July 2023). GraphPad Prism (version 8.0) was used for Friedman’s and Dunn’s multiple comparison tests to perform a pairwise comparison of Ct values. A *p*-value ≤ 0.05 was considered statistically significant.

The cross-reactivity of the MPXV RUO assays was determined by testing a pool of samples positive for the following viruses: Herpes simplex virus 1 (HSV1), Herpes simplex virus 2 (HSV2) and Varicella zoster virus (VZV). The pool was composed of positive clinical skin lesion swabs from different patients.

## 3. Results

In order to evaluate the diagnostic application of these above-described RUO assays for mpox diagnosis, 27 positive and 10 negative human skin lesions were examined.

### 3.1. Diagnostic Performance of four RUO MPXV Molecular Assays

Overall, compared with the in-house RT-PCR, true positives for MPXV were identified in all four RUO methods, except for one sample with a Ct value of 36. This latter sample was deemed a false negative in the STANDARD M10 MPX/OPX and RealCycler MONK-UX/MONK-GX v.2. For these two assays, we estimated an accuracy of 97.3% (95% CI: 86.2–99.5%), sensitivity of 96.3% (95% CI: 81.72–99.34%) and Cohen’s Kappa of 0.93 (95% CI: 0.61–1). The Novaplex MPXV Assay and RealStar Orthopoxvirus PCR Kit 1.0 achieved 100% accuracy and sensitivity with a Cohen’s Kappa of 1.00. Positive and negative predictive values ranged from 100% (95% CI: 87.1–100%) for the Novaplex MPXV Assay and RealStar Orthopoxvirus PCR Kit 1.0 to 90.9% (95% CI: 62.2–98.4%) for the STANDARD M10 MPX/OPX and RealCycler MONK-UX/MONK-GX v.2. Finally, the diagnostic specificity was tested on only 10 samples and was 100% for all MPXV RUO assays. Diagnostic performance data are summarized in Table 1 and Table 2.

### 3.2. Cross-Reactivity Assessment

To determine the analytical specificity, we tested a multi-target pool of non-MPXV pathogens using all these RUO molecular assays. This panel was characterized by a high load (Ct < 25) of HSV1, HSV2 and VZV DNA. We verified that the HSV1/2 and VZV pools were negative in all MPXV PCR kits, as confirmed by the reference method. HSV1, HSV2 and VZV DNA did not interfere with the molecular detection of MPXV in these RUO kits.

### 3.3. Comparison of Cycle Threshold Values

As shown in Figure 1, the in-house and RUO molecular tests showed different Ct distributions for high (Ct ≤ 30) and low (31 < Ct > 35) viral load specimens (*p* < 0.0001). However, the interpretation of qualitative results as weakly positive or positive was similar for all tests used, except for some low viral load samples that were characterized by a CT value of <30 using RealCycler MONK-UX/MONK-GX v.2 (N = 7 samples, excluding false negatives; Dunn’s multiple comparison test in-house vs. RealCycler *p* < 0.0001), Novaplex MPXV assay (N = 2 samples) and RealStar Orthopoxvirus PCR Kit 1.0 (N = 2 samples) and STANDARD M10 MPX/OPX (N = 2 samples) (Figure 1, panel B). Ct values of high and low positive samples are reported in the Appendix A.

### 3.4. Clinical Summary of Patients

At the time, most patients presented with cutaneous/mucosal manifestations. The morphology, size and distribution of the lesions were heterogeneous. The most common clinical presentation at diagnosis was erythematous papules or vesicles less than 10 in number (64.3%), and these lesions were located in the anogenital area. The lesions tended to erode and develop into ulcers and crusts within a few days. Lesions were frequently associated with local and systemic symptoms, namely pruritus (21.4%) or anorectal pain (35.7%), fever (35.7%) and inguinal/cervical lymphadenopathy (28.6%). The prodromal signs/symptoms, including fever (35.7%), asthenia (21.4%) and lymphadenopathy (28.6%) appeared in most MPXV-positive patients (78.6%). Only three positive patients were hospitalized for the treatment of pain and concomitant sexually transmitted infections, one of which received specific antiviral treatment. The overall outcome was favorable, with depressed skin scars at the sites of previous lesions already present at the first follow-up visit after diagnosis (21.4%). Table 3 shows the clinical data of the mpox cases. The clinical characteristics and outcomes of these trial patients have already been studied in more detail [17].

## 4. Discussion

The most common laboratory test used for the diagnosis of mpox is real-time PCR, which offers the rapid, sensitive and specific detection of the viral target, as reported in a recent review that included a total of 83 studies on natural infections of MPXV in humans. Indeed, the majority of studies included in that review reported the use of real-time PCR to provide a laboratory confirmation of mpox disease (*n* = 48) [18]. Therefore, the ongoing global surveillance activities for mpox diagnosis require the rapid provision of authorized in vitro diagnostic medical device molecular assays. Our study compared the diagnostic performance of four novel molecular kits, namely the Novaplex MPXV Assay, the STANDARD M10 MPX/OPX, the RealCycler MONK-UX/MONK-GX v.2 and the RealStar Orthopoxvirus PCR Kit 1.0, with an in-house PCR assay. Accuracy, diagnostic sensitivity and specificity were 100% for the Novaplex MPXV Assay and RealStar Orthopoxvirus PCR Kit 1.0. The RealCycler MONK-UX/MONK-GX v.2 and STANDARD M10 MPX/OPX showed a slight decrease in the accuracy and diagnostic sensitivity, which were estimated at 97.3% and 96.3%, respectively. One false negative result, defined as a low viral load specimen from the anal lesion, was observed in the RealCycler MONK-UX/MONK-GX v.2 and STANDARD M10 MPX/OPX assays. In addition, this mpox case with low viral DNA amplicons had less than 10 lesions and reported no skin/mucosal and systemic symptoms, hospitalization and administration of antiviral therapy. In view of this, in specimens with Ct ≤ 35, all RUO assays achieved an excellent diagnostic sensitivity of 100%.

Our results are consistent with the on-label analytical performance declared by the manufacturer in the instructions for the use of all the RUO assays studied. Furthermore, considering that this study included symptomatic subjects with and without mpox, our findings demonstrate a real-world diagnostic application of these assays [17].

Similar studies have been carried out using several other commercially available molecular assays [19,20,21,22]. Most of these assays detect the TNF receptor gene as the viral genomic target, while some tests were designed to amplify the interferon resistance (F3L) gene and the TNF receptor gene together with the F3L gene. Recently, some of these assays received the CE-IVD certification (i.e., RealCycler MONK-UX/MONK-GX v.2; Bio-Speedy Monkeypox Virus qPCR kits and BGI Genomics Monkeypox Virus Nucleic Acid Detection Kit). In addition, only the STANDARD M10 MPX/OPX is designed to differentiate WA MPXV from CB MPXV using specific primers and probes without increasing turnaround times. This additional result helps to identify the endemic area of the outbreak. Our clinical cases belonged to the WA MPXV, as confirmed by the sequencing of the full genome of our first regional case (Clade IIb, Lineage B.1).

Given the probability of co-infections in people at a high risk of sexually transmitted infections, a highly specific assay is required. None of these RUO kits were shown to interfere with HSV1, HSV2 and VZV genomic amplicons.

Despite these assays that showed good performance, individual laboratory assessment is required to optimize the extraction procedure to be used. However, these molecular RUO tests were easy to use and quick to prepare.

Different assays are required to satisfy the requirement of all laboratories and to allow for a prompt diagnosis in different environmental, social and economic contexts. This short evaluation provides important diagnostic information for each laboratory in choosing the most appropriate test to use. This study was limited to the small sample size and, therefore, some other diagnostic aspects were not evaluated (i.e., intra- and inter-run reproducibility and the limit of detection). The first reason was the low number of free manufacturer-provided reaction tests. Indeed, a multi-target pool was prepared and processed for the cross-reactivity analyses in order to waste less reaction material available. The second reason was the low number of mpox cases, which were diagnosed in a limited time span (from July to August 2022) [17]. 

On 11 May 2023, the “Fifth Meeting of the International Health Regulations (2005) (IHR) Emergency Committee on the Multi-Country Outbreak of mpox (monkeypox)” confirmed the transition from a Public Health Emergency of International Concern to a response and control program due to the strong decline in mpox cases. However, the WHO recommends that rapid response and action should be maintained in various medical areas of interest in order to interrupt human-to-human transmission, protect vulnerable populations and minimize the zoonotic transmission of the virus. To achieve these goals, it is very important to continue mpox research activities on vaccine safety, efficacy and the duration of protection against infection, as well as on the therapeutics and diagnostics in different countries. In view of these recommendations, the provision of fast and accurate molecular assay for mpox detection remains an urgent need to prevent viral propagation. In order to carry out active mpox surveillance, it is important to improve and simplify the diagnostic activity without requiring additional staff training and laboratory equipment everywhere, from non-endemic countries to endemic countries in Africa.

## 5. Conclusions

The Novaplex MPXV Assay, the STANDARD M10 MPX/OPX, the RealCycler MONK-UX/MONK-GX v.2 and the RealStar Orthopoxvirus PCR Kit 1.0 demonstrated excellent diagnostic performance and are suitable for routine or rapid use in clinical laboratories to differentiate mpox from other rash diseases.

## Figures and Tables

**Figure 1 microorganisms-12-00664-f001:**
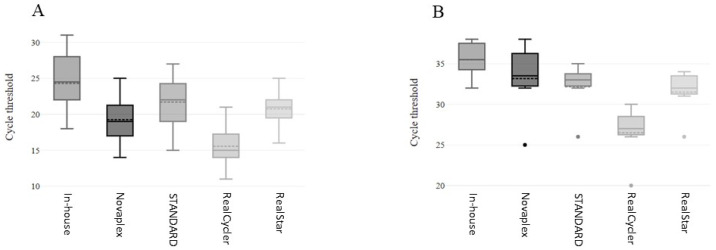
MPXV molecular diagnostics: distribution of low and high cycle threshold using in-house and research-use-only PCR assays. (**A**) Ct for high viral load specimens. (**B**) Ct for low viral load specimens. Points refer to minor values with different interpretation from the reference molecular method. Abbreviations: In-house MPXV PCR Assay (In-house); Novaplex—MPXV Assay (Novaplex); STANDARD M10 MPX/OPX (STANDARD); RealCycler MONK-UX/-GX v.2 (RealCycler); RealStar Orthopoxvirus PCR Kit 1.0 (RealStar).

**Table 1 microorganisms-12-00664-t001:** The diagnostic performance of the novel molecular MPXV assays compared to the in-house PCR procedure in use. Data are reported with 95% confidence intervals. Abbreviations: positive predictive values (PPVs) and negative predictive values (NPVs).

MPXV Real-Time PCR Assays	*Accuracy (%)*	*Sensitivity (%)*	*Specificity (%)*	*Cohen’s Kappa*	*PPV (%)*	*NPV (%)*
RealStar Orthopoxvirus PCR Kit 1.0	100 (90.6–100)	100 (87.54–100)	100 (72.25–100)	1.00 (0.67–1.00)	100 (87.5–100)	100 (72.25–100)
RealCycler MONK-UX/-GX v.2	97.3 (86.2–99.5)	96.3 (81.72–99.34)	100 (72.25–100)	0.93 (0.61–1.00)	100 (87.1–100)	90.9 (62.2–98.4)
Novaplex MPXV Assay	100 (90.6–100)	100 (87.54–100)	100 (72.25–100)	1.00 (0.67–1.00)	100 (87.54–100)	100 (72.25–100)
STANDARD M10 MPX/OPX	97.3 (86.2–99.5)	96.3 (81.72–99.34)	100 (72.25–100)	0.93 (0.61–1.00)	100 (87.1–100)	90.9 (62.2–98.4)

**Table 2 microorganisms-12-00664-t002:** Concordance among the RUO assays and the in-house PCR test for the MPXV diagnosis of clinical human skin specimens.

		In-House	
		+	−
*Novaplex MPXV Assay*	+	27	0
	−	0	10
*STANDARD M10 MPX/OPX*	+	26	0
	−	1	10
*RealCycler MONK-UX/MONK-GX v.2*	+	26	0
	−	1	10
*RealStar Orthopoxvirus PCR Kit 1.0*	+	27	0
	−	0	10

**Table 3 microorganisms-12-00664-t003:** Clinical overview and management of mpox patients.

*Number*	14
*Age (y)*	34 (IQR: 27–43)
*Male*	14 (100%)
Homosexual	13 (92.9%)
Bisexual	1 (7.1%)
*Smallpox vaccination history*	1 (7.1%)
*Other sexually transmitted diseases*	11 (78.6%)
HIV	3 (21.4%)
Syphilis	8 (57.1%)
HR-HPV	5 (35.7%)
*Ureaplasma*/*Mycoplasma* spp.	5 (35.7%)
Chlamydia trachomatis	1 (7.1%)
Neisseria gonorrhoeae	3 (21.4%)
PrEP	3 (21.4%)
*Onset of illness*	
Prodromal signs/symptoms	11 (78.6%)
Fever	5 (35.7%)
Myalgia	1 (7.1%)
Asthenia	3 (21.4%)
Headache	2 (14.3%)
Urticaria	1 (7.1%)
Mucorrhea	1 (7.1%)
Rectorrhagia/Proctorrhagia	2 (14.3%)
Ano/Rectal pain	5 (35.7%)
Itch	3 (21.4%)
Pharyngodynia	2 (14.3%)
Lymphadenopathy	4 (28.6%)
Tenesmus	1 (7.1%)
*Skin/mucosal manifestation at diagnosis*	12 (85.7%)
*Approximate number of lesions (mucosal or cutaneous)*	
<10	9 (64.3%)
10–20	3 (21.4%)
>20	2 (14.3%)
*Resolution without scarring*	12 (85.7%)
*Unconventional specimens with monkeypox viral DNA (a)*	
Blood (Whole blood, serum or plasma)	4 of 7 (57%)
Nasopharyngeal swab	6 of 10 (60%)
Rectal/Anal swab	7 of 7 (100%)
*Antiviral therapy*	1 (7.1%)
*Hospitalization*	3 (21.4%)
Days of hospital admission	4 (2–5)

Values are reported as median (interquartile range) or number (percentage). (a) The detection of monkeypox viral DNA was investigated in blood (whole blood, serum and plasma), nasopharyngeal swabs and rectal/anal swabs for patients with more than 10 lesions and illness onset with one of the systemic signs/symptoms.

## Data Availability

Data are contained within the article and Appendix A.

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
