# Peer review of "Molecular Diagnosis of Human Monkeypox Virus during 2022–23 Outbreak: Preliminary Evaluation of Novel Real-Time Qualitative PCR Assays"

_microorganisms, 2024, doi:10.3390/microorganisms12040664_

Round 1
Reviewer 1 Report
Comments and Suggestions for Authors
I consider the sample N to be low if the objective is to validate a diagnostic kit, in this sense, I suggest that the study was coherent, had good results, however, the sample N is not enough to validate a kit.
Author Response
We thank the reviewer for raising this central point of discussion for this revision. The aim of our study was to evaluate the diagnostic performance of four commercial real-time molecular assays recommended for research use only for the detection of Monkeypox virus. Clinical specimens were collected from a variety of skin/mucosal lesions, ranging from erythematous papules/vesicles/pustules to crusts and ulcers, mainly in the anogenital area. Our sample size reflects the epidemiological records of Monkeypox virus between June and August 2022 in Liguria (Italy). This study limitation was reported on line 268 in the Discussion section.
Reviewer 2 Report
Comments and Suggestions for Authors
DePace et al. present a comparison of in-house and commercial Real-Time PCR assays for detection of monkeypox virus from a small sample of positive and negative clinical samples. They find all the assays to be very accurate, with only a single sample misidentified by two of four commercial assays. This work largely confirms what has been presented in other studies. The work is for the most part clearly presented, although there are a few garbled passages that need proofreading. There seems to be one serious discrepancy, where the misidentified sample is listed as a false negative in the results, but a fals positive in the discussion. With a single discordant sample, I don't think they can really say which assays are "higher performance".
Specific items by line number:
31 Cohen’s Kappa (not k). This typo also occurs in section 3.1.
37 The disease should be referred to as mpox (formerly known as Monkeypox). The virus has not yet been officially renamed.
49 Men (and transgender women) who have sex with men (MSM) have been more affected. Whether they identify as gay, bisexual or straight does not affect risk.
59 This sentence is garbled.
74 How were these samples determined to be true positive or negative?
92 Was there some reason to target this particular gene?
170 Can’t really claim “higher” performance with only a single discordant sample (this could be statistically by chance). Should just report the sensitivity/specificity/accuracy numbers. I will defer to a real statistician about whether Cohen’s kappa is meaningful here.
200 The clinical summary is clinically interesting but not clear it is relevant to what this study is testing.
204 87.5% is not “almost all”. Can say “most” or “a large majority”.
228 Garbled sentence/paragraph
241 Wasn’t this a false NEGATIVE?
Comments on the Quality of English Language
Overall clear. Some garbled sentences that look like cut-and-paste errors.
Author Response
We appreciate this brief summary of our study. According to the reviewer, all molecular assays tested have good performances for the detection of Monkeypox virus in skin/mucosal swabs, whole blood, serum or plasma, nasopharyngeal swabs and rectal/anal swabs. As the reviewer correctly suggested, we reported the misidentified sample as a false positive in the discussion on line 238. We corrected this typographical error.
Specific items by line number:
- 31 Cohen’s Kappa (not k). This typo also occurs in section 3.1.
- The disease should be referred to as mpox (formerly known as Monkeypox). The virus has not yet been officially renamed.
- 49 Men (and transgender women) who have sex with men (MSM) have been more affected. Whether they identify as gay, bisexual or straight does not affect risk.
- 59 This sentence is garbled.
- 74 How were these samples determined to be true positive or negative?
- 92 Was there some reason to target this particular gene?
- 170 Can’t really claim “higher” performance with only a single discordant sample (this could be statistically by chance). Should just report the sensitivity/specificity/accuracy numbers. I will defer to a real statistician about whether Cohen’s kappa is meaningful here.
- 200 The clinical summary is clinically interesting but not clear it is relevant to what this study is testing.
- 204 87.5% is not “almost all”. Can say “most” or “a large majority”.
- 228 Garbled sentence/paragraph
- 241 Wasn’t this a false NEGATIVE?
- Overall clear. Some garbled sentences that look like cut-and-paste errors.
AR
We thank the reviewer for these corrections. We have revised the manuscript as recommended.
The following questions are answered in detail:
- 49 Men (and transgender women) who have sex with men (MSM) have been more affected. Whether they identify as gay, bisexual or straight does not affect risk.
We appreciate your question about the risk category of the population for mpox infection. Gay, bisexual and other men who have sex with men are the main victims of a global monkeypox outbreak in 2022. However, we have reported that it is sexual behaviour that increases exposure to infection, not identification as gay, bisexual or other men who have sex with men (as shown in lines 51-53). We provide some references:
Suárez Rodríguez B, Guzmán Herrador BR, Díaz Franco A, Sánchez-Seco Fariñas MP, Del Amo Valero J, Aginagalde Llorente AH, et al. Epidemiologic features and control measures during monkeypox outbreak, Spain, June 2022. Emerg Infect Dis. 2022;28(9):1847-51.
Ciccarese, G.; Di Biagio, A.; Bruzzone, B.; Guadagno, A.; Taramasso, L.; Oddenino, G.; Brucci, G; Labate, L.; De Pace, V.; Mastrolonardo, M. et al. Monkeypox outbreak in Genoa, Italy: Clinical, laboratory, histopathologic features, management, and outcome of the infected patients. J. Med. Virol. 2023, 95, e28560.
- 74 How were these samples determined to be true positive or negative?
True positive and negative results were defined using a validated and standardized in-house multiplex real-time PCR as reported in lines 88-108. In addition, a clinical positive and negative control was used when amplifying a target sequence. This is to confirm that the primer-probe set is working.
- 92 Was there some reason to target this particular gene?
MPXV generic G2R_G was validated with high reaction efficiency by Li Y. et al. (2010). We provide the reference:
Li Y, Zhao H, Wilkins K, Hughes C, Damon IK. Real-time PCR assays for the specific detection of monkeypox virus West African and Congo Basin strain DNA. J Virol Methods. 2010 Oct;169(1):223-7. doi: 10.1016/j.jviromet.2010.07.012. Epub 2010 Jul 17. PMID: 20643162; PMCID: PMC9628942.
Reviewer 3 Report
Comments and Suggestions for Authors
The study by de Pace and colleagues evaluated four commercially available and one previously published in-house test for the molecular detection of monkeypox virus. Given the recent appearance of the disease in non-endemic countries, it is important for such comparisons and validations of all available putative diagnostics tools to occur, so that laboratories worldwide are prepared for future outbreaks. The study is well-designed.
Comments on the Quality of English LanguageI would only suggest some minor corrections in syntax from a native speaker.
Author Response
We thank the reviewer for this positive note. About fine/minor spell check English language and style as evidenced by the Reviewers, manuscript was revised from a native speaker English.